# Monitoring *Chilecomadia valdiviana* (Lepidoptera: Cossidae) Using Sex Pheromone-Baited Traps in Apple Orchards in Chile

**DOI:** 10.3390/insects12060511

**Published:** 2021-06-01

**Authors:** Wilson Barros-Parada, Eduardo Fuentes-Contreras, Jan Bergmann, Heidy Herrera, Takeshi Kinsho, Yuki Miyake

**Affiliations:** 1Escuela de Agronomía, Pontificia Universidad Católica de Valparaíso, Casilla 4-D, Quillota 2260000, Chile; 2Centro de Ecología Molecular y Funcional, Facultad de Ciencias Agrarias, Universidad de Talca, Casilla 747, Talca 3460000, Chile; efuentes@utalca.cl; 3Instituto de Química, Pontificia Universidad Católica de Valparaíso, Valparaíso 2340000, Chile; jan.bergmann@pucv.cl; 4Núcleo de Química y Bioquímica, Facultad de Ciencias, Universidad Mayor, La Pirámide 5750, Santiago 8580745, Chile; heidy.herrera@mayor.cl; 5Shin-Etsu Chemical Co. Ltd., 6-1, Ohtemachi 2-chome, Chiyoda-ku, Tokyo 100-0004, Japan; kinsho@shinetsu.jp (T.K.); miyake_y@shinetsu.jp (Y.M.)

**Keywords:** carpenterworm, male attractant, monitoring, flight phenology, (7*Z*,10*Z*)-7,10-hexadecadienal, wood-borer

## Abstract

**Simple Summary:**

A series of experiments were carried out to evaluate the efficacy of trap type, sex pheromone (Z7,Z10-16:Ald) dose, and trap height on attraction to males of *Chilecomadia valdiviana* (Philippi) (Lepidoptera: Cossidae) in the canopy of apple orchards in Chile. Based on trap catches and relative costs, we propose that bucket traps (6 L) baited with 300 μg of sex pheromone, regardless of canopy height, are suitable for male mass trapping, while delta traps baited with 300 μg of sex pheromone are adequate for monitoring of male flight.

**Abstract:**

*Chilecomadia valdiviana* (Philippi) (Lepidoptera: Cossidae) is a native xylophagous pest in apple orchards in Chile. A series of experiments evaluated the efficacy of trap type, sex pheromone (Z7,Z10-16:Ald) dose, and trap location in the apple tree canopy on trap catch of male adults. Bucket traps (6 L), with and without roof and cross vane spacers, together with bucket traps (20 L) without roof and spacers, showed higher catches among the four types of traps evaluated. In a second experiment, the UNI-trap and Delta trap showed higher catches than Multipher, wing, and bucket traps (6 L). Male catches were not affected by height when tested at 0, 1.5, and 3 m in the canopy. A 300 µg dose of Z7,Z10-16:Ald showed higher catch than the control treatment. This dose allowed monitoring of male flight of *C. valdiviana* for at least five weeks in apple orchards in Chile. Based on relative trap costs, we propose the use of 6 L bucket traps for male mass trapping, while Delta traps can be used for monitoring of male flight. We found that male flight of *C. valdiviana* occurred mainly from mid-August to late November, reaching its maximum in mid-September.

## 1. Introduction

*Chilecomadia valdiviana* (Philippi) (Lepidoptera: Cossidae) is a xylophagous native moth, distributed in the subantartic forests of the Andes mountains [1]. This carpenterworm causes economic damage in fruit orchards (e.g., apple, pear, cherry, avocado) and forest plantations (poplar and eucalyptus) in the south-central zone of Chile [2,3,4,5,6,7]. The damage is caused by the larval stages when they feed on the wood, forming galleries that progressively weaken the host plant [6]. The adult flight period, evaluated with emergence traps and non-specific light traps, occurs over a period of 150–210 days in spring and summer [8,9]. Owing to the lack of effective control methods [10], fruit growers who face this problem are forced to remove any branches or trees that show high levels of damage, repeating this process annually. As larvae are located inside the wood, and adults are the main life stage targeted by control measures, good estimation of the flight phenology could improve management of this pest.

The blend of compounds in the female pheromone has been identified by Herrera et al. [11]. The major pheromone component is (7*Z*,10*Z*)-7,10-hexadecadienal (Z7,Z10-16:Ald) and the minor compounds are (*Z*)-7-hexadecenal, (*Z*)-9-hexadecenal, hexadecanal, and (9*Z*,12*Z*)-9,12-octadecadienal [11]. Later studies demonstrated that the minor compounds do not enhance long-range male attraction over Z7,Z10-16:Ald alone [12]. Knowledge of the pheromone of this moth opens possibilities for its sustainable management through flight monitoring, mating disruption, or mass trapping [13,14,15,16,17].

Efficient monitoring of moth pests [13,17], including the estimation of their flight phenology and timing of control measures [18,19], is possible if both lure and trap design can be optimized with field tests. Variables to consider for trap optimization include trap type [20], color [21,22,23,24], size [25], quality of the adhesive surface [26], height of trap [25,27], and relative position of the trap in the canopy [25], and factors to consider for lure optimization include type of attractant [28], attractant dose [29], lure type [30,31,32,33], and lure position in the trap [34].

Specifically for cossid moths, several studies have evaluated the importance of various trap–lure combinations for monitoring and mass trapping. Cylinder and diamond traps showed higher catches than screen or board traps for *Prionoxystus robiniae* (Peck) in the USA [35,36], with different types of lures (hollow plastic fiber, rubber septum, and cotton wick) showing differences in catches and attraction period, depending on the region and the moth population density [36,37,38]. For this species, higher catches were associated with higher pheromone concentration or emission rates for several types of lures [36]. The same was also true for the species *Acossus centerensis* (Lintner) [39]. Furthermore, higher catches were found higher in the canopy for *P. robiniae*, with no effect of trap color [37], and the same was also true for *A. centerensis* across various lures and trap heights [39]. For *Coryphodema tristis* (Drury) in eucalyptus plantations in South Africa, cross vane bucket traps performed better than bucket funnel or Delta traps [40]. Similarly, the color of plastic funnel traps had no effect on catches of *Cossus cossus* L. in *Populus* plantations in Italy [41]. Ardeh et al. [42] found that doubling the size of a trapezoidal trap (and hence doubling the adhesive area), along with a greater entrance area, doubled the catch of moths compared with the standard delta traps for *Zeuzera pyrina* (L.) in walnut orchards in Iran. Once adequate trap–lure combinations were recognized, their use allowed the dynamics of adult flight of *A. centerensis* [39] and *Z. pyrina* [43,44] to be determined.

Here, we report the results of experiments conducted within apple orchards in the Maule Region (Chile), evaluating the efficacy of the following variables on trap catch of males of *C. valdiviana*: (i) trap type, (ii) trap height in the tree canopy, and (iii) dose of Z7,Z10-16:Ald in rubber septa. Furthermore, we determined the seasonal dynamics of male flight in apple orchards.

## 2. Materials and Methods

### 2.1. Preparation of Lures

The pheromone lures for the trap type and trap height experiments were prepared by loading white rubber septa (Sigma-Aldrich, catalog #Z553905) with 100 μL of a hexane solution containing 300 µg of synthetic Z7,Z10-16:Ald, which was synthesized by Shin-Etzu Chimical Co, Ltd. (Tokyo, Japan), as described previously in Lapointe et al. [12].

### 2.2. Exp. #1: Efficacy of Custom-Made Bucket Traps

Experiment #1 was run in Orchard 1 located in Colbún, Chile (35°43′44″ S, 71°27′12″ W). We used 2.5 ha of a 22-year-old apple orchard (cv. Royal Gala), planted with a spacing of 4.5 × 2.2 m (1010 trees/ha), trained in central axis, and with an average height of 4 m. To compare the captures of male *C. valdiviana,* four types of custom-made white bucket traps were used as follows: (i) 20 L with roof and intercept cross-vane spacers, (ii) 6 L with roof and intercept cross-vane spacers, (iii) 20 L without roof or intercept cross-vane spacers, and (iv) 6 L without roof or intercept cross-vane spacers. These white bucket traps were baited with the pheromone lures, which were fixed to the trap with a pin under the roof (in treatments i and ii) or in a hole in a 2 × 5 cm card holder (in treatments iii and iv), and a control treatment without any lure (white bucket 20 L without roof or intercept cross-vane spacers). All traps were filled with a solution of 300 g of sodium chloride and 5 mL of liquid glycerin soap in 1.25 L of water [11]. For all treatments, there were five replicates. The traps were positioned within the canopy at a height of 2.8 m, oriented to the southwest with no leaves covering the trap entrance. Traps were placed in every fourth row, spaced every 12 trees, for a separation of 26.4 × 27 m. Traps were installed in the field on 11 October and checked weekly until 25 October 2018.

### 2.3. Exp. #2. Efficacy of Commercial Traps

Experiment #2 was also conducted in Orchard 1. Captures of males of *C. valdiviana* were compared using four types of commercial traps: (i) UNI-trap yellow/white (Alphascent, West Linn, OR, USA), (ii) Multipher I green/white (Solida, Saint-Ferréol-les-Neiges, QC, Canada), (iii) white Delta trap (Alphascent, Oregon, USA), and (iv) white wing trap (Alphascent, Oregon, USA), as well as (v) the custom-made white bucket trap (6 L) without roof or intercept cross-vane spacers, filled with a saline solution. Each trap was baited with a pheromone lure, which was placed inside the basket of the trap (treatments i and ii), or fixed with a pin under the trap’s roof (treatments iii and iv), or in a card holder, as previously described in Exp. #1 (treatment v). For all treatments, there were five replicates, with the same trap location and spacing as described for Exp. #1. Traps were placed in the field on 5 October and checked weekly and rotated on each sampling date until 14 November 2018.

### 2.4. Exp. #3. Efficacy of Trap Height

Experiment #3 was also conducted in Orchard 1. To compare captures of male *C. valdiviana* moths at different canopy heights, the following three heights were evaluated: (i) 0 m (ground level), (ii) 1.5 m, and (iii) 3 m above ground in the tree canopy. We used white bucket 6 L traps filled with a saline solution and baited with a lure fixed with a card holder as previously described in Exp. #1 (treatment v). For all treatments, there were four replicates, with the same trap location and spacing pattern as in Exp. #1. Traps were placed in the field on 11 October and checked weekly until 25 October 2016.

### 2.5. Experiment #4. Efficacy of Pheromone Dose

Experiment #4 was carried out in Orchard 2, located in Colbún, Chile (35°43′53″ S, 71°27′43″ W). We used 2.8 ha of 22-year-old apple trees (cv. Royal Gala), planted with a spacing of 4.5 × 2.5 m (889 trees/ha), trained in central axis, and with an average height of 3.6 m. To compare the captures of male *C. valdiviana* moths, lures were prepared by loading them with different doses of Z7,Z10-16:Ald as follows: (i) 3, (ii) 10, (iii) 30, (iv) 100, and (v) 300 μg per lure, plus a control containing only hexane. The trap used was the white bucket 6 L trap filled with a saline solution and baited with one of the lures, which was fixed with a card holder as previously described in Exp. #1 (treatment v). For all treatments, there were six replicates. The traps were located within the canopy at a height of 2.0 m, oriented to the southwest with no leaves blocking the trap entrance. Traps were placed in the field on 7 October and checked weekly and rotated on each sampling date until 25 December 2015. The last moths were caught on 7 December 2015.

### 2.6. Seasonal Moth Flight

Our flight phenology study was conducted in three managed apple orchards in the Maule Region in Chile. Orchards 1 and 2 are described above, while Orchard 3 was located in Colbún, Chile (35°43′10.44″ S, 71°27′42.42″ W). We used 6.5 ha of this 23-year-old apple orchard planted with ‘Royal Gala’ (33.3%) and ‘Scarlet’ (66.7%) cultivars, with a spacing of 4 × 1.8 m (1389 trees/ha), trained with central axis, and having an average height of 3.8 m. The trap used was the white 20 L bucket trap filled with a saline solution and baited with a lure, which was fixed with a card holder as previously described in Exp. #1 (treatment v). The pheromone lure was changed every eight weeks until the end of the moth flight season. The traps were located within the canopy at height of 3 m, oriented to the southwest with no leaves blocking the trap entrance, with at least 100 m of separation between the continuously running traps within each plot. Traps were installed on 7 August 2016 and checked weekly until 3 March 2017.

### 2.7. Statistical Analysis

For all experiments, differences among treatments were determined for male captures per trap per day (or week). As the data distributions did not fulfill the assumption of homoscedasticity, the non-parametric Kruskal–Wallis multiple comparison test with Bonferroni correction was used [45,46].

## 3. Results

### 3.1. Exp. #1. Efficacy of Custom-Made Bucket Traps 

In Exp. #1, 949 males of *C. valdiviana* were captured in a two-week period. The four types of bucket traps caught more male moths than the control bucket trap without the pheromone lure (Table 1). During the second week, and for the mean catch of the whole period, the white bucket 20L trap with roof and cross vane spacers captured significantly fewer male moths than the other treatments baited with a pheromone lure (Table 1).

### 3.2. Exp. #2. Efficacy of Commercial Traps

A total of 1186 males of *C. valdiviana* were captured in five weeks, with significant differences between trap types during the first three weeks of the experiment (Table 2). Mean captures for the whole sampling period were significantly higher for UNI-trap and Delta traps in relation with other trap types, while the Wing trap showed significantly higher captures than Multipher I and Bucket traps (Table 2).

### 3.3. Exp. #3. Efficacy of Trap Height 

We captured 238 males of *C. valdiviana* in the two weeks of this experiment. There were no significant differences in captures of males of *C. valdiviana* between different trap heights in the tree canopy (Table 3).

### 3.4. Exp. #4. Efficacy of Lure Pheromone Concentration

We captured 722 males of *C. valdiviana* in the 15 weeks of this experiment. The highest dose of 300 µg Z7,Z10-16:Ald caught more moths than the control treatment during at least four weeks (14 September–25 October) (Table 4). All the other doses of Z7,Z10-16:Ald had catches similar to the control treatment on all dates. The mean values of the whole experiment showed the same pattern (Table 4).

### 3.5. Seasonal Moth Flight

A total of 5596 males of *C. valdiviana* were captured in the three orchards over the full season of monitoring male moth flight. The first males appeared between 12 and 19 August 2016 (Figure 1). Subsequently, male catch increased steadily towards the end of August, peaking between 21 and 27 September 2016. Captures decreased after 15 November, until nearly ending on 28 January 2016 (Figure 1).

## 4. Discussion

Trap type has an important influence on male catch of cossid moths [35,36], with larger traps or traps with greater adhesive surface catching more male moths [42]. In our experiments, Uni-traps and Delta traps caught more moths than Multipher, wing, or bucket traps. Delta traps seem to be a suitable alternative for monitoring male flight, with weekly removal of the sticky liners during the maximum catch period (September–October). On the other hand, for mass trapping, Uni-traps can catch more males than Multipher or bucket traps. Although bucket traps (6L) caught fewer male moths, their cost is ca. 2.5% of the cost of Uni-traps and Multipher traps. All traps tested collected male moths over several weeks with little servicing required. We found that roof and cross vane spacers did not increase the catch in bucket traps, which differs from the results reported by Bouwer et al. [40]. Therefore, we recommend that growers use the cheaper bucket traps (6L) without roof or cross vane spacers for mass trapping of *C. valdiviana* in Chilean apple orchards.

Optimizing trap height in the canopy increased the catch of *P. robiniae* [2,3,4,5,6,7] and *A. centerensis* [39] in hardwood forest in the USA. In our field experiment, trap height did not affect trap catch, which suggests that male flight is not restricted in terms of height in apple orchards. As apple trees have a lower canopy height (4 m) than the types of hardwood forest trees previously evaluated (10–15 m) [37,39], we found this variable is not important for *C. valdiviana* male catch in apple orchards in Chile.

Traps baited with lures containing 300 μg of Z7,Z10-16:Ald caught more moths than the control treatments within a period of five weeks. This duration is enough time to cover the main period of male moth flight (September–October) in Chile. *Chilecomadia valdiviana* is the first species of Cossidae known to have an aldehyde pheromone [11], which is susceptible to degradation under field conditions [47]. However, we found that replacement of the lures loaded with 300 μg pheromone every eight weeks (56 days) allowed us to monitor the flight of males of *C. valdiviana* under field conditions during the whole season. This is consistent with the monitoring and mass trapping protocols reported for *Z. pyrina* in olive orchards, which suggest replacing pheromone lures every 40 days, although a much higher pheromone dose of 10 mg applied in polyethylene bags was used in these studies [43,44]. The higher dose might have been necessary for the type of dispenser used, because, presumably, the diffusion rate through the walls of the polyethylene bags is lower compared with the release rate of the pheromone from rubber septa. We found that the flight period of *C. valdiviana* males in apple orchards started around mid-August with a single flight maximum in late September. In this period, apple orchards are in full bloom in Chile and pollinator activity precludes the use of non-selective insecticides. Therefore, control of males could be achieved using sexual pheromone-based technologies during this period. Our studies indicate that Delta traps could be used for early adult male monitoring and detection of infested areas, because they should be inspected regularly and the change of sticky liners prevents trap saturation. On the other hand, white bucket traps (6L) could be used for control of male moths with mass trapping techniques. Although the cost of mass trapping will depend on the number of traps per unit of area to be protected, catches of the bucket trap (6L) were approximately 40% of the catches of UNI-trap. This reduction in trap catch could be compensated with an increase in the number of traps per unit of area. Further studies on trap density and combination with other attractive kairomone compounds or UV light stimuli to achieve bisexual monitoring or mass trapping [43], as well as the evaluation of the effect of mass trapping during several seasons on tree damage, could contribute to the sustainable management of *C. valdiviana* in apple orchards in Chile.

## 5. Conclusions

Bucket traps (6 L) with and without roof and cross vane spacers together with bucket traps (20 L) without roof or cross vane spacers showed higher catches of *C. valdiviana* among the four type of traps evaluated.UNI-traps and Delta traps had higher catch of *C. valdiviana* than Multipher, wing, or bucket traps.Male catch was not affected by height when tested at 0, 1.5, and 3.0 m in the canopy.The 300 µg dose of Z7,Z10-16:Ald caught more males of *C. valdiviana* than the control treatment.Male flight of *C. valdiviana* occurred mainly from mid-August to late November, reaching their maximum in mid-September in apple orchards in Chile.

## Figures and Tables

**Figure 1 insects-12-00511-f001:**
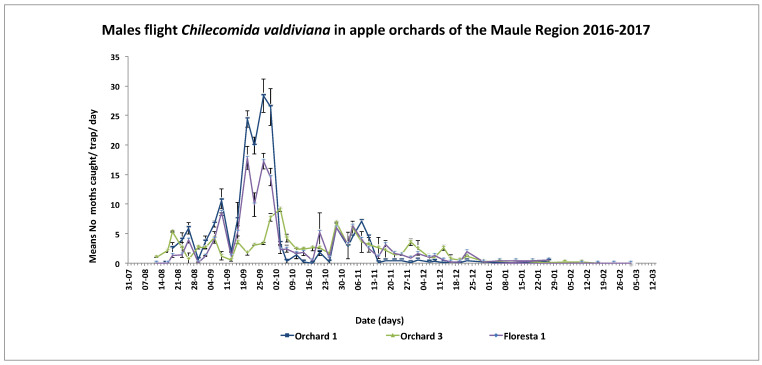
Seasonal flight activity of *Chilecomadia valdiviana* males monitored with sex pheromone baited bucket traps at three apple orchards in 2016–2017 in the Maule Region, Chile.

**Table 1 insects-12-00511-t001:** Captures of male *Chilecomadia valdiviana* moths in different types of white bucket traps in an apple orchard during the 2016 season in the Maule Region, Chile.

Treatment	No. of Males Trap^−1^ Day^−1^ (Mean (SE))		
	Week 1 (Oct 18)	Week 2 (Oct 25)	Mean
(i) 20 L with roof and cross vane spacers	2.14 ^a^ (0.54)	3.29 ^b^ (1.02)	2.72 ^b^ (1.97)
(ii) 20 L without roof or cross vane spacers	2.50 ^a^ (0.46)	5.79 ^a^ (0.85)	4.14 ^a^ (2.86)
(iii) 6 L with roof and cross vane spacers	2.54 ^a^ (1.54)	7.21 ^a^ (1.93)	4.88 ^a^ (1.66)
(iv) 6 L without roof or cross vane spacers	3.54 ^a^ (1.01)	6.89 ^a^ (1.42)	5.21 ^a^ (1.70)
Control (no pheromone lure) 20 L without roof or cross vane spacers	0 ^b^	0 ^c^	0 ^c^
	H_(4,3)_ = 10.7	H_(4,3)_ = 11.7	H_(4,3)_ = 11.7
	*p* < 0.05	*p* < 0.05	*p* < 0.05

^a,b,c^ Values followed by different letters in the same column are different according to Kruskal–Wallis multiple comparison test with Bonferroni correction. *n* = 4.

**Table 2 insects-12-00511-t002:** Captures of male *Chilecomadia valdiviana* in different types of traps in an apple orchard during 2016 in the Maule Region, Chile.

Treatment	No. of Males Trap^−1^ Day^−1^ (Mean (SE))					
	Week 1 (Oct 16)	Week 2 (Oct 23)	Week 3 (Oct 30)	Week 4 (Nov 05)	Week 5 (Nov 13)	Mean
UNI-trap (i)	4.15 ^a^ (0.50)	1.89 ^a^ (0.32)	1.06 ^b^ (0.10)	0.70 ^a^ (0.14)	0.23 ^a^ (0.05)	1.60 ^a^ (0.16)
Multipher I trap (ii)	1.16 ^c^ (0.24)	0.66 ^d^ (0.13)	0.74 ^d^ (0.08)	0.63 ^a^ (0.10)	0.28 ^a^ (0.05)	0.69 ^c^ (0.05)
Delta trap (iii)	3.38 ^a^ (0.20)	1.49 ^b^ (0.22)	1.37 ^a^ (0.11)	0.70 ^a^ (0.14)	0.28 ^a^ (0.05)	1.44 ^a^ (0.06)
Wing trap (iv)	2.07 ^b^ (0.24)	1.51 ^b^ (0.31)	0.91 ^c^ (0.12)	0.80 ^a^ (0.21)	0.23 ^a^ (0.05)	1.11 ^b^ (0.14)
Bucket trap (v)	0.65 ^d^ (0.17)	1.06 ^c^ (0.23)	0.91 ^c^ (0.26)	0.57 ^a^ (0.17)	0.25 ^a^ (0.04)	0.69 ^c^ (0.10)
	H_(4,4)_ = 21.0	H_(4,4)_ = 9.9	H_(4,4)_ = 10.0	H_(4,4)_ = 1.2	H_(4,4)_ = 1.4	H_(4,4)_ = 19.3
	*p* < 0.05	*p* < 0.05	*p* < 0.05	*p* = 0.88	*p* = 0.84	*p* < 0.05

^a,b,c^ Values followed by different letters in the same column are different according to Kruskal–Wallis multiple comparison test with Bonferroni correction. *n* = 5.

**Table 3 insects-12-00511-t003:** Captures of male *Chilecomadia valdiviana* in traps at different heights in the tree canopy in an apple orchard during 2016 in the Maule Region, Chile.

Height (m)	No. of Males Trap^−1^ Day^−1^ (Mean (SE))		
	Week 1 (Oct 18)	Week 2 (Oct 25)	Mean
0	1.25 ^a^ (0.44)	1.71 ^a^ (0.23)	1.48 ^a^ (0.3)
1.5	0.57 ^a^ (0.23)	1.82 ^a^ (0.19)	1.20 ^a^ (0.18)
3.0	1.21 ^a^ (0.24)	1.93 ^a^ (0.32)	1.57 ^a^ (0.24)
	H_(2,3)_ = 3.0 *p* = 0.22	H_(2,3)_ = 0.9 *p* = 0.95	H_(2,3)_ = 1.3 *p* = 0.52

^a^ Values followed by different letters in the same column are different according to Kruskal–Wallis multiple comparison test with Bonferroni correction. *n* = 4.

**Table 4 insects-12-00511-t004:** Captures of male *Chilecomadia valdiviana* in traps baited with different doses of the main component Z7,Z10-16:Ald of the sexual pheromone in an apple orchard during 2016 in the Maule Region, Chile.

Dose (µg)	No. of Males Trap^−1^ Day^−1^ (Mean (SE))													
	14 Sep	21 Sep	28 Sep	06 Oct	12 Oct	18 Oct	25 Oct	02 Nov	08 Nov	15 Nov	22 Nov	29 Nov	07 Dec	Mean
300	1.14 ^a^ (0.27)	0.86 ^a^ (0.17)	0.88 ^a^ (0.31)	2.92 ^a^ (0.28)	1.81 ^a^ (0.18)	0.08 ^a^ (0.04)	0.40 ^a^ (0.15)	0.40 ^a^ (0.10)	0.42 ^a^ (0.19)	0.24 ^a^ (0.16)	0.12 ^a^ (0.09)	0.10 ^a^ (0.07)	0.29 ^a^ (0.12)	0.74 ^a^ (0.23)
100	0.62 ^a^ (0.22)	0.48 ^ab^ (0.10)	0.60 ^a^ (0.20)	1.29 ^ab^ (0.25)	0.67 ^ab^ (0.15)	0.00 ^a^ (0.00)	0.12 ^ab^ (0.06)	0.04 ^a^ (0.03)	0.06 ^a^ (0.06)	0.05 ^a^ (0.05)	0.00 ^a^ (0.00)	0.00 ^a^ (0.00)	0.00 ^a^ (0.00)	0.30 ^ab^ (0.11)
30	0.86 ^a^ (0.12)	0.14 ^ab^ (0.04)	0.33 ^a^ (0.09)	0.17 ^ab^ (0.04)	0.14 ^ab^ (0.08)	0.03 ^a^ (0.03)	0.00 ^b^ (0.00)	0.00 ^a^ (0.00)	0.00 ^a^ (0.00)	0.05 ^a^ (0.05)	0.00 ^a^ (0.00)	0.00 ^a^ (0.00)	0.00 ^a^ (0.00)	0.13 ^ab^ (0.07)
10	0.64 ^a^ (0.20)	0.02 ^ab^ (0.02)	0.07 ^a^ (0.05)	0.00 ^b^ (0.00)	0.00 ^b^ (0.00)	0.00 ^a^ (0.00)	0.00 ^b^ (0.00)	0.00 ^a^ (0.00)	0.00 ^a^ (0.00)	0.00 ^a^ (0.00)	0.00 ^a^ (0.00)	0.00 ^a^ (0.00)	0.00 ^a^ (0.00)	0.06 ^ab^ (0.05)
3	0.21 ^a^ (0.12)	0.02 ^ab^ (0.02)	0.07 ^a^ (0.03)	0.00 ^b^ (0.00)	0.00 ^b^ (0.00)	0.00 ^a^ (0.00)	0.00 ^b^ (0.00)	0.00 ^a^ (0.00)	0.00 ^a^ (0.00)	0.02 ^a^ (0.02)	0.00 ^a^ (0.00)	0.00 ^a^ (0.00)	0.00 ^a^ (0.00)	0.03 ^b^ (0.02)
0	0.52 ^a^ (0.18)	0.00 ^b^ (0.00)	0.02 ^a^ (0.02)	0.00 ^b^ (0.00)	0.00 ^b^ (0.00)	0.00 ^a^ (0.00)	0.00 ^b^ (0.00)	0.00 ^a^ (0.00)	0.00 ^a^ (0.00)	0.00 ^a^ (0.00)	0.00 ^a^ (0.00)	0.00 ^a^ (0.00)	0.00 ^a^ (0.00)	0.04 ^b^ (0.04)
H_(5,36)_	9.9	29.5	15.5	33.2	31.5	12.0	27.5	23.1	12.6	7.2	10.3	10.3	18.0	30.1
*p*	0.079	0.001	0.008	0.001	0.001	0.034	0.001	0.003	0.028	0.205	0.068	0.068	0.003	0.001

^a,b,c^ Values followed by different letters in the same column are different according to Kruskal–Wallis multiple comparison test with Bonferroni correction. *n* = 6.

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
