# Peer review of "Monitoring Chilecomadia valdiviana (Lepidoptera: Cossidae) Using Sex Pheromone-Baited Traps in Apple Orchards in Chile"

_insects, 2021, doi:10.3390/insects12060511_

Round 1

Reviewer 1 Report

The study evaluated the efficacy of sex pheromone trap designs (types), deployment time and position, and pheromone doses in capturing male moths of the lepidopteran pest Chilecomadia valdiviana of apple trees in Chile.  The study design appears reasonable and sound and results are supported by data.  However, there are some major issues on data analyses that need to be addressed, particularly in the analysis of data from the pheromon experiment:

(1) The authors used non-parametric (Kruskal-Wallis) ANOVA and mean comparison procedures for all day analyses because of the violation of equal variance among different treatments.  That is okay for the one-factor experimental design.  However, all the one-factor experiment had to deal with the time factor - i.e., catches of the moths at different times.  It is not clear how the authors dealt with the time factor - as repeated measures or simply lumped them together as total captures. 

(2)  It's not appropriate to conduct multiple comparisons among different doses in any dose-response study.  An appropriate  regression approach should be used for Experiment 4.

Minor edits:

Title  remove "of" after "monitoring"

Line 2:  Change "effect" to "efficacy"

Remove "effect" from all subheadings as they are repetitive.

Author Response

Response Reviewer #1:

General comments

The study evaluated the efficacy of sex pheromone trap designs (types), deployment time and position, and pheromone doses in capturing male moths of the lepidopteran pest Chilecomadia valdiviana of apple trees in Chile. The study design appears reasonable and sound and results are supported by data. However, there are some major issues on data analyses that need to be addressed, particularly in the analysis of data from the pheromon experiment:

Comment 1. “The authors used non-parametric (Kruskal-Wallis) ANOVA and mean comparison procedures for all day analyses because of the violation of equal variance among different treatments. That is okay for the one-factor experimental design. However, all the one-factor experiment had to deal with the time factor - i.e., catches of the moths at different times. It is not clear how the authors dealt with the time factor - as repeated measures or simply lumped them together as total captures”.

We agree with Reviewer 1 that we use a no-parametric ANOVA analysis (Wruskal-Walis) that leaves out time. We think that since a two-way ANOVA is not possible because it violates the assumptions of variance, the non-parametric analysis of the average of the catches offers a reproducible and possible solution for this group of authors.

Comment 2. ”It's not appropriate to conduct multiple comparisons among different doses in any dose-response study. An appropriate regression approach should be used for Experiment 4”.

We tried to perform a linear regression. However, the linear regression does not comply with the assumptions of Homoscedasticity and Normality of the residuals, we attempted to transform the data, but these presented atypical values. We performed the non-parametric Kruskal-Wallis test as an alternative to the situation according to what was used by Knight et al. 2010 for 8-17 week trials.

Reviewer 2 Report

This is a well written report of experiments conducted to assess the trap type, location in canopy, and pheromone amount in catching male C. valdiviana in apple orchards in Chile.  It is a short article that provides practical information about the use of sex pheromone traps for monitoring the pest in orchards.  The last sentence of the discussion indicates that mass trapping could be a way to control this pest but it is uncertain of the pest density in orchards.  A statement about pest density in the apple orchards used in these studies would be helpful. Mass trapping might work if the pest density is low.  I just have a few specific comments listed below.

Line 112- give the full name and location of Ch-V lures.  I am not familiar with this company. Is this a type of lure supplied by Shin-Etsu?  Was the aldehyde synthesized in house or by another company?

Tables should include horizontal dividers to better determine which values belong to which treatments.

Figure 1 – y-axis label has typos.  Change to ‘Mean # moths caught/trap/day’

The x-axis can be labeled date instead of time.

Author Response

Reviewer #2:

General comments

This is a well written report of experiments conducted to assess the trap type, location in canopy, and pheromone amount in catching male C. valdiviana in apple orchards in Chile.  It is a short article that provides practical information about the use of sex pheromone traps for monitoring the pest in orchards.  The last sentence of the discussion indicates that mass trapping could be a way to control this pest but it is uncertain of the pest density in orchards.  A statement about pest density in the apple orchards used in these studies would be helpful. Mass trapping might work if the pest density is low.  I just have a few specific comments listed below.

We value the comments of reviewer 1, tending to improve our work with their contributions. According to the discussion, we limit the general comments of the manuscript at the end and address them with a better explanation of the scope of our study. The areas evaluated generally correspond to sectors with 2-4 active galleries per tree and 25-38% more infested trees (n = 3 Tree infected, presence of new and old galleries, presence of sawdust and spots). We fully agree where mass trapping is recommended early and at low levels of infestation. Mass trapping has a good chance of being used in the sustainable management of this species.

Specific comment

Line 114 Reviewer 2 points us: give the full name and location of Ch-V lures.  I am not familiar with this company. Is this a type of lure supplied by Shin-Etsu?  Was the aldehyde synthesized in house or by another company?

We accepted the comment reviewers #1 and clarify that the pheromone was provided by Shin-Etsu and we remove the name Ch-V as follows “The pheromone lures for the trap type and trap height experiments were prepared by loading white rubber septa (Sigma-Aldrich, catalog #Z553905) with 100 μL of a hexane solution containing 300 μg of synthetic Z7,Z10-16:Ald (Ch-V lures), which was … synthesized by Shin-Etsu Chemical Co, Ltd (Tokyo, Japan),… as described previously in Lapointe et al. [12].”

Lines 118, 131, 151, 170, 201, 202, 222 and 226. We eliminate the name Ch-V that we gave to the bait in the whole manuscript.

Example “These white bucket traps were baited with Ch-V the pheromone… lures, which were fixed to the trap(…)”

Reviewer 2 points us: Tables should include horizontal dividers to better determine which values belong to which treatments.

Table 1, 2, 3, and 4. We accept the comments of reviewer #1 and add horizontal dividers to make the easier to read treatments.

Reviewer 2 points us: Figure 1 – y-axis label has typos.  Change to ‘Mean # moths caught/trap/day’

Figure 1. We accept the comments of reviewer #1 and make the change in Y-axis.

Reviewer 2 points us : Figure 1 The x-axis can be labeled date instead of time.

Figure 1. We accept the comments of reviewer #1 and make the change in X-axis.

Reviewer 3 Report

This manuscript presents a comparison of the effectiveness of various models of pheromone traps intended to capture males of the moth Chilecomadia valdiviana. I only have a few points of detail to improve the impact. 

Giving the same name “effect of trap”, to exp 1 and 2 is confusing.

Line 78 there are several studies that have evaluated …

Line 82 dispensers ?

Line 180 also

Table 2 I understand you omitted the letters for statistical tests in col for weeks 4 and  5 because of non significance but at first glance I looked for it. Also headings for col 5  and col mean should be in bold characters.

Idem for Table 3. If you do not put letters, then change the legend

Table 4 contains many numbers and would be advantageously replaced by a figure. The main information: level of catches and dispenser life-time are both positively correlated to doses would be easier to catch on a graph.

Line 255 English. The type of trap has an important influence

Discussion:

Line 269 In view of the very short distance, it is not certain that the catch rate is correlated with the height where the males have the greatest probability of finding a female. Many other explanations could be offered.

Line 283 – Not sure one can compare the raw amounts within dispensers. The diffusion rate of polyethylene bags, which involves permeation through walls, is probably lower compared to evaporation from the surface of rubber septum.

Line 288 – This conclusion should be better supported. A priori, for all types of use, the trap that captures the most should be retained. It is not clear to me what are the specific arguments that allow one to favor a particular type of trap. For example, the total cost depends on the number of traps per unit area to be protected, a delta trap is more easily saturated with large insects and requires more maintenance ...

Author Response

Reviewer #3:

General comments

This manuscript presents a comparison of the effectiveness of various models of pheromone traps intended to capture males of the moth Chilecomadia valdiviana. I only have a few points of detail to improve the impact.

Specific comments

Reviewer 3 points us : Giving the same name “effect of trap”, to exp 1 and 2 is confusing.

Line 121 and Line 219 Exp 1: Effect of trap type (custom-made bucket traps)

Exp. #1: Effect of Custom-made Bucket Traps

Line 143 and line 231 Exp 2: Effect of trap type (commercial traps)

Exp. #2. Effect of Commercial Traps

Reviewer 3 points us : Line 80 there are several studies that have evaluated … We accepted and made the change

Reviewer 3 points us : Line 82 dispensers ? We eliminate dispensers

Reviewer 3 points us : Line 195 also We accepted the change and eliminate also

Reviewer 3 points us : Table 2 I understand you omitted the letters for statistical tests in col for weeks 4 and  5 because of non significance but at first glance I looked for it. Also headings for col 5  and col mean should be in bold characters.

Reviewer 3 points us : Idem for Table 3. If you do not put letters, then change the legend

In the Table 2, 3, and 4 We accept the comments and add the same letters in the treatments that do not have significant differences.

Reviewer 3 points us : Table 4 contains many numbers and would be advantageously replaced by a figure. The main information: level of catches and dispenser life-time are both positively correlated to doses would be easier to catch on a graph.

In the Table 4 We think that a graph has the disadvantage of losing the averages of the dates in the last column. So we added the letters in the table as he suggested.

Discussion (The numbers correspond to the version previously sent)

Reviewer 3 points us : Line 269 (Corresponds to the line of the previous document) In view of the very short distance, it is not certain that the catch rate is correlated with the height where the males have the greatest probability of finding a female. Many other explanations could be offered.

We accept the comments of reviewer #2 and make the change “In our field experiment, trap height did not affect trap catch, which suggests …that male flight is not restricted in terms of height in apple orchards. Since apple trees… have lower canopy height (4 m) than the types of hardwood forest trees previously evaluated (10-15 m) [37, 39], we found this variable is not important for C. valdiviana male catch in apple orchards in Chile.”

Reviewer 3 points us : Line 283 (Corresponds to the line of the previous document) – Not sure one can compare the raw amounts within dispensers. The diffusion rate of polyethylene bags, which involves permeation through walls, is probably lower compared to evaporation from the surface of rubber septum.

We accept the comments of reviewer #2 and make the change “(…)although a much higher pheromone dose of 10 mg applied in polyethylene bags was used in these studies [43-44]. …The higher dose might have been necessary for the type of dispenser used, because presumably the diffusion rate through the walls of the polyethylene bags is lower than compared to the release rate of the pheromone from rubber septa…”

Reviewer 3 points us : Line 288 (Corresponds to the line of the previous document) – This conclusion should be better supported. A priori, for all types of use, the trap that captures the most should be retained. It is not clear to me what are the specific arguments that allow one to favor a particular type of trap. For example, the total cost depends on the number of traps per unit area to be protected, a delta trap is more easily saturated with large insects and requires more maintenance ...

We accept the comments of reviewer #2 and make the change “Our studies indicate that delta traps could be used for early adult male monitoring and detection of infested areas, …because they should be inspected regularly and the change of sticky liners prevent trap saturation. On the other hand,… bucket traps (6L) could be used for control of male moths with mass trapping techniques. …Although the cost of mass trapping will depend on the number of traps per unit of area to be protected, catches of the bucket trap (6L) were approx. 40% of the catches of UNI-trap. This reduction in trap catch could be compensated with an increase in the number of traps per unit of area”...